# Students' preparedness for disasters in schools: a systematic review protocol

Hamed Seddighi,[1,2] Homeira Sajjadi,[3] Sepideh Yousefzadeh,[4] Mónica López López,[5] Meroe Vameghi,[6] Hassan Rafiey,[2,6] Hamid Reza Khankeh,[7] Magdalena Garzon Fonseca [8]

► Additional material is published online only. To view please visit the journal online (http://dx.doi.org/10.1136/bmjpo-2020-000913).

For numbered affiliations see end of article.

**Correspondence to**
PhD Candidate Magdalena Garzon Fonseca; m.garzon. fonseca@rug.nl

PhD Candidate Hamed Seddighi, Student Research Committee, University of Social Welfare and Rehabilitation Sciences, Tehran, Iran; hseddighi@gmail.com

Dr. Homeira Sajjadi; Safaneh_ s@yahoo.com

## ABSTRACT

**Introduction** Children are one of the most vulnerable groups in disasters. Improving students' knowledge and skills to prepare for disasters can play a major role in children's health. School as a place to teach children can make a significant contribution to provide the necessary skills. This study aims to identify the effects, strengths and weaknesses of interventions in schools to prepare children for disasters.

**Methods and analysis** We use Preferred Reporting Items for Systematic Reviews and Meta-Analyses guidelines to develop a protocol for this systematic review. The included studies will report on the results of interventions targeting 'schoolchildren' defined as individuals between 4 and under 18 years old studying in schools. Different electronic databases will be used for a comprehensive literature search, including MEDLINE, Web of Science, CINAHL, PsycINFO, Cochrane Register of Controlled Trials and EMBASE to identify the records that match the mentioned inclusion criteria published till December 2020. The main search terms are 'disaster', 'preparedness', 'children' and 'school'. Four types of data will be extracted from the qualified studies including study characteristics (study design, year of publication and geographical region where the study was conducted), participant characteristics (sample size, age and gender), intervention characteristics (aim of intervention, intervention facilitators and barriers) and intervention outcomes. The quality appraisal of the selected papers will be conducted using Cochrane Collaboration's Risk of Bias for quantitative studies and Critical Appraisal Skills Programme checklist for qualitative studies. We use a narrative synthesis for this systematic review. The narrative synthesis refers to an approach to systematic reviews which focuses mostly on applying words and texts to summarise and explain findings.

**Ethics and dissemination** This paper is a part of a Ph.D. thesis of Hamed Seddighi at University of Social welfare and Rehabilitation Sciences with ethics code IR.USWR.REC.1399.008 approved by the Ethics Committee of the above-mentioned university.

**PROSPERO registration number** CRD42020146536.

## INTRODUCTION

In recent decades, more disasters have occurred.[1] Disasters are one of the major threats to children's health.[2] According to the origin, disasters triggered by natural hazards and technological hazards. Natural hazards

### What is known about the subject?

► In previous studies were indicated that preparing children could be useful and effective (eg, evacuation drills in Chile, preparing students for hurricane in Cuba and tsunami preparedness in Japan). Innovative solutions using digital tools for preparing children for climate-related disasters are effective.
► Preparing marginalised school students for disasters is vital. Intersection of childhood and racial and ethnic social class, disability, gender and residence inequalities increases vulnerability and therefore increases disasters' risk for children.

### What this study hopes to add?

► This study provides a comprehensive summary of studies investigating school-based education programmes for preparing children for disasters.
► Provide useful information for policy makers, governments, researchers and humanitarian organisations about state of art of programmes for preparing children for disasters around the world.
► Highlight closing the gaps for preparing children for disasters such as earthquake, flood and pandemics.

are categorised as geophysical (earthquakes, landslides, tsunamis and volcanic activity), hydrological (avalanches and floods), climatological (extreme temperatures, drought and wildfires), meteorological (cyclones and storms/wave surges) or biological (disease epidemics and insect/animal plagues).[3] Technological hazards are such as conflicts, famine, displaced populations, industrial accidents.[3] WHO defined 'All-hazard approach' as a concept acknowledging that, while hazards vary in source (natural, technological, societal), they often challenge health systems in similar ways. Thus, risk reduction, emergency preparedness, response actions and community recovery activities are usually implemented along the same model, regardless of the cause'.[4] Children are one of the most vulnerable groups during and after disasters.[5]

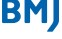

Disasters can seriously damage children's physical and mental health.[6] It also harms children indirectly, so that its outcomes may last a lifetime.[7 8] In addition, disasters destroy children's educational infrastructure and disrupt learning in children.[9] Moreover, following disasters, violence against children increases in sexual, physical, psychological and neglect forms.[10] Child trafficking has been reported as one of the impacts of disasters in the affected areas.[11] Children who have lost their parents or caregivers and the children whose parents cannot afford to live will have to work after disasters.[12] Early marriage is another case reported in the affected areas, especially for girls.[13 14] Each of the above-mentioned cases also affects social, economic and political structures.[15] These consequences will last for generations.[16] Although timely response and good governance have a significant effect in preventing such consequences, children's readiness can be one of the main solutions to reduce postdisaster losses.[17 18]

In previous decades, most managers and decision-makers would think that children are not able to take care of themselves and they need help from adults.[19] Therefore, most interventions for disaster preparedness were defined for adults. But now researchers have found that children can respond appropriately to an accident and stay healthy.[20] The United Nations has recognised raising the children's awareness as one of the main pillars of children's readiness.[20]

School as a place where almost all children gather to enjoy the opportunity to learn, can be the best place to prepare them for disasters.[21] The required knowledge and skill can be provided to the child in various ways in schools, one of which is practical training and skills development.[22] Although various studies have been conducted on the interventions in schools to prepare the children for disasters, no systematic review has been yet conducted to identify methods and challenges.

## Aim of review

The main purpose of this study is to identify the methods, outcomes, facilitators and barriers of educational programmes in schools to prepare children for disasters. In addition, the differences of processes and outcomes in terms of type of disasters and social inequalities, including geographical location, age, gender, disability, ethnic and religion belief, are investigated. However, as the fundamental aim of the school-based programmes is to prepare children in order to reduce trauma in those exposed. It is not possible for the proposed narrative synthesis to address the effectiveness of these programmes in reducing trauma among exposed children.

## METHOD

We use Preferred Reporting Items for Systematic Reviews and Meta-Analyses guidelines to develop the protocol for this systematic review (online supplemental material 1).[23] In addition, Cochrane guidance is applied in the development of a systematic review protocol to specify the research questions using Population, Interventions, Comparators and Outcomes elements. Studies will be assessed against clearly defined criteria to determine their inclusion or exclusion in the review. Finally, the findings of the included studies will be assessed and reported.[24]

## Eligibility
### Inclusion criteria
#### Type of participants
The included studies will report on the results of interventions targeting 'school children' defined as individuals under 18 years old studying in schools.

#### Context/setting
The age of compulsory school entry is various in the world but it is normally 4–7 years old. For this review, we will consider the children aged 4–18 in public or private schools.

#### Type of interventions
The included studies will report on the interventions delivered in schools with a focus on disasters' preparedness education. Various interventions with different duration, timing, and modality will be acceptable. Acceptable interventions are those delivered in full or in part in the schools. Such interventions were delivered by teachers or other people playing a disaster preparedness role. Schools are chosen as the primary setting because many programmes for children are delivered in schools (ie, shakeout drills).

#### Type of studies
The studies included in this review will be as the following: qualitative studies (those evaluating the process of school-based educational interventions for preparedness against disasters), observational studies (including case–control, cohort and cross-sectional studies) and clinical trials (including non-randomised and randomised controlled trials) examining the effect of disaster preparedness education interventions on school students.

#### Types of publication
The study will include empirical research works published in peer-reviewed journals or conference proceedings that are accompanied with full-length peer-reviewed papers.

#### Outcome measures
The included studies will report on the interventions the primary outcome of which is an appropriate focus on the children's preparedness (knowledge, attitude and behaviour) for disasters.

The primary goal of all included interventions is to improve children preparedness for disasters. For this reason, included studies shall report on at least one educational instrument applied to measure children's preparedness.

## Exclusion criteria
### Type of participants
The study will exclude individuals under 4 and above 18 years old.

### Context/setting
Studies will be excluded if they report interventions delivered in settings other than the schools.

### Type of interventions:
Studies will be excluded if (1) they do not include a disasters' preparedness intervention for improving skills and changing knowledge, attitude and behaviour, (2) they report on the interventions delivered without a school-based component (ie, disasters' preparedness education for children via television.)

### Type of studies
Studies will be excluded if they are not original intervention studies, they are published in journals that are not peer-reviewed. They will be excluded if they do not report on a disasters' preparedness.

## Search strategy
We will use electronic databases for comprehensive literature search including MEDLINE, Web of Science, CINAHL, PsycINFO, Cochrane Register of Controlled Trials (CENTRAL) and EMBASE to identify the records meeting the mentioned inclusion criteria and published till April 2020. Different keywords for the systematic search were identified during the initial literature search. The main search terms are 'disaster', 'preparedness', 'children' and 'school'. The combinations of keywords related to population, interest and context and the draft of MEDLINE search strategy is presented in online supplemental material 2. Search terms will be combined with the appropriate Boolean operators and the search will be based on titles, abstracts and keywords.

## Selection processes
Two reviewers will participate in the selection process. An Endnote desktop will be used to store references and subsequently identify and remove duplicates. For finding eligible studies, abstracts and full texts will be reviewed. Two reviewers will then separately scan abstracts and full texts of currently eligible studies against the eligibility requirements of the research, taking into consideration the type of intervention, sample population and the recorded outcomes. The two reviewers must come to an agreement through discussion and if they fail to reach a compromise, a third reviewer may have a resolution.

## Data extraction
Two reviewers shall individually perform data retrieval, and agreement will be achieved through conversation. The following data will be extracted from the qualified studies:
► Study characteristics: study design, year of publication and geographical location of study conduct.

► Participant characteristics: sample size, age (eg, mean with SD, range) and gender.
► Intervention characteristics: aim of intervention, intervention facilitators and barriers.
► Intervention outcome.

## Quality appraisal
The quality appraisal of the selected papers will be conducted separately by two reviewers using Cochrane Collaboration's Risk of Bias (CCRB).[25 26] CCRB will be used for measuring risk of bias for selected articles with randomised design. Bias is surveyed in five dimensions, including selection bias, performance bias, detection bias, attrition bias and reporting bias.

For assessing risk of bias in qualitative studies, we will use the Critical Appraisal Skills Programme checklist recommended by the Cochrane Collaboration for qualitative literature.[27] This tool has 10 questions (nine questions addressing quality and one addressing 'value').

## Data synthesis
Due to the heterogeneity and variation of the studies to be reviewed, especially wide range of disasters, we will not use a statistical aggregation of the data. Instead, a narrative synthesis will be used for this systematic review. In the narrative data synthesis, we will use from 'all hazard approach' that was defined in the introduction. The narrative synthesis refers to an approach to systematic reviews that focuses mostly on use of words and texts to summarise and explain findings.[28] It is usually considered as the 'second best' approach for synthesis in systematic reviews. In fact, this approach is a significant method to interpret findings extensively used in policy and practice. This approach will discuss on the effects of interventions and factors shaping the implementation of interventions. According to Popay et al, in order to conduct the narrative synthesis, four steps shall be followed[28]:
► Developing a theory of how the intervention works, why and for whom.
► Developing a preliminary synthesis of findings of included studies.
► Exploring relationships in the data.
► Assessing the robustness of the synthesis.

The first step will contribute to the interpretation of the study findings and to assess how widely the findings may be applicable. The second step is developing a preliminary synthesis. In this step, the aim is to identify and list facilitators and barriers to implementation of interventions on children's preparedness in schools for disasters. Exploring relationships in the data will be the third step. This step aims to explain the facilitators and/or barriers to successful implementation across the included studies and to understand how and why the interventions have an effect. In the final step, the robustness of the synthesis product will be assessed. The aim of this step is to provide an assessment of the strength of the evidence in order to draw conclusions about the facilitators and/or barriers to implementation identified in the

synthesis. For developing a preliminary synthesis, three methods will be used including textual descriptions, grouping and clusters, and tabulation.

In textual descriptions, a descriptive paragraph on the included studies will be presented. In grouping method, we aim to cluster findings to aid the process of analysing and finding patterns. Studies will be categorised according to the type of interventions, contexts and outcomes. In addition, tabulation will be used to summarise some findings.

## CONCLUSIONS

In the planned systematic review, we aim to identify methods, outcomes, facilitators and barriers of school-based education programmes for preparing children for disasters. Such a review can raise decision-makers' knowledge of disaster preparedness for children. Recognising best practices will be an opportunity for countries researching for interventions. Researchers can also find solutions to remove the barriers to preparedness of children for disasters.

**Author affiliations**
[1]Student Research Committee, University of Social Welfare and Rehabilitation Sciences, Tehran, Iran
[2]Department of Social Welfare, University of Social Welfare and Rehabilitation Sciences, Tehran, Iran
[3]Social Determinants of Health Research Center, University of Social Welfare and Rehabilitation Sciences, Tehran, Iran
[4]University Campus Fryslân, University of Groningen, Leeuwarden, Friesland, Netherlands
[5]Faculty of Behavioural and Social Sciences, University of Groningen, Groningen, Netherlands
[6]Social Welfare Management Research Center, University of Social Welfare and Rehabilitation Sciences, Tehran, Iran
[7]Health in Emergency and Disaster Research Center, University of Social Welfare and Rehabilitation Sciences, Tehran, Iran
[8]Department of Child & Family Welfare, Graduate School of Behavioral Sciences, University of Groningen, Groningen, Netherlands

**Contributors** HSe is the guarantor. HSe drafted the manuscript. HSa, SY, HR, MV, HRK and MGF provided edits for manuscript improvement. MV developed the search strategy. HSa, SY and ML contributed to the revision of the report. All authors reviewed, contributed to and approved the final version of the manuscript. All authors have read and agreed to the published version of the manuscript.

**Funding** The authors have not declared a specific grant for this research from any funding agency in the public, commercial or not-for-profit sectors.

**Competing interests** None declared.

**Patient and public involvement statement** Patients and public were not involved in the development of this protocol.

**Patient consent for publication** Not required.

**Ethics approval** This paper is a part of a Ph.D. thesis of Hamed Seddighi at University of Social welfare and Rehabilitation Sciences with ethics code IR.USWR. REC.1399.008 approved by the Ethics Committee of the mentioned university.

**Provenance and peer review** Not commissioned; externally peer reviewed.

**Data availability statement** Data are available on reasonable request.

**ORCID iD**
Magdalena Garzon Fonseca http://orcid.org/0000-0002-6984-9577

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
