## [Reviewer comments · BMJ Paediatrics Open]

ARTICLE DETAILS

TITLE (PROVISIONAL)	Students' Preparedness for Disasters in Schools: A Systematic Review Protocol
AUTHORS	Seddighi, Hamed Sajjadi, Homeira Yousefzadeh, Sepideh Lopez, Monica Vameghi, Meroe Rafiey, Hassan Khankeh, Hamid Reza Garzon Fonseca, Magdalena

VERSION 1 – REVIEW

REVIEWER	Reviewer name: Dr. Luis Rajmil Institution and Country: Homer 22 1rst 1, Barcelona, 08023, Spain Competing interests: None
REVIEW RETURNED	09-Nov-2020

GENERAL COMMENTS	This protocol try to answer an important question regarding to what extent schools conduct interventions to prepare children for natural disasters. The protocol, searching and quality criteria looks appropriate. Authors would be taken into account minor changes to try to improve the protocol: 1) Regarding the exclusion criteria, the protocol explicitly excludes studies that analyze human-made disasters (i.e. nuclear accidents, mass shooting, industrial accident and the like). However, apart from some of the examples that are proposed, there is growing evidence that many natural disasters are related to climate change, and that climate change is produced directly or indirectly by man. For this reason it is proposed to mention the impact of climate change and deepen the differentiation between natural events and secondary events to human activity. 2) Another aspect to study is social inequalities. Natural disasters could have the greatest impact on the most vulnerable populations with the fewest resources, both between countries and within each country. And these inequalities could be increased if the most vulnerable receive less training and interventions: inverse care law. Authors could add a specific objective in order to assess social inequalities
--

REVIEWER	Reviewer name: Dr. Nick Spencer Institution and Country: University of Warwick Warwick Medical School, Division of Mental Health and Wellbeing, United Kingdom of Great Britain and Northern Ireland Competing interests: None
REVIEW RETURNED	10-Nov-2020

GENERAL COMMENTS	This systematic review addresses a challenging research question. The review has received approval from PROSPERO and
--

	makes appropriate use of established guidelines for conducting robust reviews. Children are undoubtedly highly vulnerable to natural hazards and, if school-based preparedness interventions can be shown to work, they might contribute to reducing the physical and mental trauma associated with experiencing these hazards. The review aims to identify the effects, facilitators and barriers of educational programmes in schools to prepare children for natural hazards. The basic methodology of the review is sound; however, I have a number of suggestions which I think will strengthen the presentation of this review protocol in the paper:  1. Databases and search strategy:  a. The exclusion of unpublished findings (e.g. PhD theses) may introduce publication bias and the authors should justify this decision b. Development of a search strategy is a skilled task usually requiring a professional librarian. The authors should specify the professional qualification of the search strategy developer/s 2. Natural hazards listed in the search strategy: The natural hazards considered are wide ranging and very different in their type of effects (e.g. floods v. Ebola). The authors should specify how these differences will be addressed in the narrative synthesis 3. Exclusions: Page 7 lines 15-16 states, among criteria for exclusion, “ when the intervention results and outcomes do not appear to have an appropriate effect on preparedness”. This reads as if publications with negative findings will be excluded. The authors should clarify this statement. 4. The effects of the programmes: The fundamental aim of the programmes under review is to prepare children in order to reduce trauma in those exposed. It is not possible for the proposed narrative synthesis to address the effectiveness of these programmes in reducing trauma among exposed children and the authors should state this clearly. Specifically, they need to clarify what they mean by ‘effects’ in the section on Aims of the Review (pp. 42-43).
--	--

VERSION 1 – AUTHOR RESPONSE

List of revision in the text to address Reviewers’ Comments

Journal: BMJ Paediatrics Open

Manuscript ID: bmjpo-2020-000913

Title: " Students’ Preparedness for Disasters in Schools: A Systematic Review Protocol"

The authors acknowledge reviewers for their valuable comments on the manuscript. Addressing these comments certainly improved paper and I now hope the paper is acceptable for publication in BMJ Paediatrics Open. The comments of the reviewers and the associated action from authors are described here.

Reviewer: 1

1) Regarding the exclusion criteria, the protocol explicitly excludes studies that analyze human-made disasters (i.e. nuclear accidents, mass shooting, industrial accident and the like). However, apart from some of the examples that are proposed, there is growing evidence that many natural disasters are related to climate change, and that climate change is produced directly or indirectly by man. For this reason, it is proposed to mention the impact of climate change and deepen the differentiation between natural events and secondary events to human activity.

Action Taken:

Thank you for your valuable comment. The manuscript was modified and “natural disasters” term was replaced with “disasters” term. As you truly mentioned, disasters are not natural and produced directly or indirectly by human nature.

Reviewer: 1

2) Another aspect to study is social inequalities. Natural disasters could have the greatest impact on the most vulnerable populations with the fewest resources, both between countries and within each country. And these inequalities could be increased if the most vulnerable receive less training and interventions: inverse care law. Authors could add a specific objective in order to assess social inequalities

Action Taken:

Thank you very much for your valuable comment. The manuscript was modified according to your valuable suggestion as follows:

The main purpose of this study is to identify ... the differences of processes and outcomes in terms of type of disasters and social inequalities including geographical location, age, gender, disability, ethnic, and religion belief are investigated.

Reviewer: 2

1. Databases and search strategy:

a. The exclusion of unpublished findings (e.g. PhD theses) may introduce publication bias and the authors should justify this decision

Action Taken:

Thank you for your valuable comment. It was deleted this exclusion criterion.

Reviewer: 2

b. Development of a search strategy is a skilled task usually requiring a professional librarian. The authors should specify the professional qualification of the search strategy developer/s.

Action Taken:

Thank you for your valuable comment. Dr. Meroe Vameghi has professional qualification of search strategy because of her academic background. It was noted in the “Contributors” section that she developed search strategy.

Reviewer: 2

2. Natural hazards listed in the search strategy:

The natural hazards considered are wide ranging and very different in their type of effects (e.g. floods v. Ebola). The authors should specify how these differences will be addressed in the narrative synthesis.

Action Taken:

Thank you for your valuable comment. It was specified as follows:

Due to the heterogeneity and variation of the studies to be reviewed, especially wide range of natural hazards, we will not use a statistical aggregation of the data. Instead, a narrative synthesis. In the narrative data synthesis, we will use from "all hazard approach". With definition of the World Health Organization, "All-hazard is a concept acknowledging that, while hazards vary in source (natural, technological, societal), they often challenge health systems in similar ways. Thus, risk reduction, emergency preparedness, response actions and community recovery activities are usually implemented along the same model, regardless of the cause".

Reviewer: 2

3. Exclusions:

Page 7 lines 15-16 states, among criteria for exclusion, “when the intervention results and outcomes

do not appear to have an appropriate effect on preparedness". This reads as if publications with negative findings will be excluded. The authors should clarify this statement.

Action Taken:

Thank you for your valuable comment. The third exclusion criteria was deleted.

Reviewer: 2

4. The effects of the programmes:

The fundamental aim of the programmes under review is to prepare children in order to reduce trauma in those exposed. It is not possible for the proposed narrative synthesis to address the effectiveness of these programmes in reducing trauma among exposed children and the authors should state this clearly. Specifically, they need to clarify what they mean by 'effects' in the section on Aims of the Review (pp. 42-43).

Action Taken:

Thank you for your valuable comment. It was added sentences to the "aim section" of the protocol according to your valuable suggestion as follows:

However, as the fundamental aim of the school-based programs is to prepare children in order to reduce trauma in those exposed. It is not possible for the proposed narrative synthesis to address the effectiveness of these programs in reducing trauma among exposed children

Editor in Chief

Comments to the Author:

What is already known Replace existing statements with examples from the recent review article (Seddighi H, Yousefzadeh S, López López M, et al Preparing children for climate-related disasters BMJ Paediatrics Open 2020;4:e000833. doi: 10.1136/bmjpo-2020-000833)eg benefit of school practice for hurricane.

Action Taken:

Thank you for your valuable comment. The What is already known section was modified according to your valuable suggestion with examples from our recent article.

Editor in Chief

What this study hopes to add delete "First study that provides"

Action Taken:

Thank you for your valuable comment. It was deleted.

Editor in Chief

Do not exclude non-English studies. Far better to try and include all. Those studies that you cannot translate can be excluded in your analysis

Action Taken:

Thank you for your valuable comment. It was modified the exclusion criteria according to your valuable suggestion and we will not exclude non-English studies.